# Comparing animal well-being between bile duct ligation models

**Guanglin Tang**[1,2☯], **Wiebke-Felicitas Nierath**[1☯], **Emily Leitner**[1☯], **Wentao Xie**[1], **Denis Revskij**[3], **Nico Seume**[1], **Xianbin Zhang**[1,4], **Luise Ehlers**[2], **Brigitte Vollmar**[1], **Dietmar Zechner**[1]*

**1** Rudolf-Zenker-Institute for Experimental Surgery, Rostock University Medical Center, Rostock, Germany, **2** Department of General Surgery, Fushun Central Hospital, Fushun, Liaoning, China, **3** Division of Gastroenterology, Department of Medicine II, Rostock University Medical Center, Rostock, Germany, **4** Department of General Surgery & Institute of Precision Diagnosis and Treatment of Gastrointestinal Tumors, Shenzhen University General Hospital & Shenzhen University Clinical Medical Academy, Shenzhen, China

☯ These authors contributed equally to this work.
* dietmar.zechner@uni-rostock.de

**Data Availability Statement:** The datasets used and/or analyzed during the current study can be downloaded from figshare: https://doi.org/10.6084/m9.figshare.25752060.

## Abstract

A prevailing animal model currently used to study severe human diseases like obstructive cholestasis, primary biliary or sclerosing cholangitis, biliary atresia, and acute liver injury is the common bile duct ligation (cBDL). Modifications of this model include ligation of the left hepatic bile duct (pBDL) or ligation of the left bile duct with the corresponding left hepatic artery (pBDL+pAL). Both modifications induce cholestasis only in the left liver lobe. After induction of total or partial cholestasis in mice, the well-being of these animals was evaluated by assessing burrowing behavior, body weight, and a distress score. To compare the pathological features of these animal models, plasma levels of liver enzymes, bile acids, bilirubin, and within the liver tissue, necrosis, fibrosis, inflammation, as well as expression of genes involved in the synthesis or transport of bile acids were assessed. The survival rate of the animals and their well-being was comparable between pBDL+pAL and pBDL. However, surgical intervention by pBDL+pAL caused confluent necrosis and collagen depositions at the edge of necrotic tissue, whereas pBDL caused focal necrosis and fibrosis in between portal areas. Interestingly, pBDL animals had a higher survival rate and their well-being was significantly improved compared to cBDL animals. On day 14 after cBDL liver aspartate, as well as alanine aminotransferase, alkaline phosphatase, glutamate dehydrogenase, bile acids, and bilirubin were significantly elevated, but only glutamate dehydrogenase activity was increased after pBDL. Thus, pBDL may be primarily used to evaluate local features such as inflammation and fibrosis or regulation of genes involved in bile acid synthesis or transport but does not allow to study all systemic features of cholestasis. The pBDL model also has the advantage that fewer mice are needed, because of its high survival rate, and that the well-being of the animals is improved compared to the cBDL animal model.

**Funding:** This study received support by the Deutsche Forschungsgemeinschaft (DFG research group FOR 2591, ZE 712/1-1, ZE 712/1-2, VO 450/15-1 and VO 450/15-2). LE and DR were funded by the research project "EnErGie" from the European Social Fund (ESF; reference: ESF/14-BM-A55-007/18) and the Ministry of Education, Science, and Culture of Mecklenburg-Vorpommern. The funders had no role in study design, data collection and analysis, decision to publish, or preparation of the manuscript.

**Competing interests:** The authors have declared that no competing interests exist.

## Introduction

The common bile duct ligation (cBDL) is a well-established animal model for severe human diseases like obstructive cholestasis [1, 2], primary biliary or sclerosing cholangitis [1, 3], biliary atresia [1], and acute liver injury [2–4]. Cholestatic conditions cause elevated levels of bile salts, bilirubin, and plasma activity of liver enzymes such as aspartate aminotransferase (AST), alanine aminotransferase (ALT), alkaline phosphatase (ALP), as well as glutamate dehydrogenase (GLDH) in the blood and induce typical local features of liver injury such as, necrosis, inflammation, and fibrosis [2–6].

Necrotic areas can be noticed already on day 1 after cBDL, and remain to be seen for several weeks [4]. Hepatic infiltration of neutrophils is a rapid response to liver injury [7, 8]. Studies have demonstrated that neutrophil granulocytes accumulate following bile duct ligation, exacerbate liver damage [4, 9], and persist at elevated levels even 14 days after induction of cholestasis [10]. On day 14 or later time points after cBDL fibrosis can be observed [4, 11, 12]. Cholestasis also often regulates the expression of genes involved in bile acid synthesis or the transportation of bile acids in form of an adaptive response. For example, cBDL represses the expression of three key enzymes *Cyp7a1* [13–15], *Cyp8b1* [15, 16], and *Cyp27a* [15, 17], involved in the novo synthesis of bile acids from cholesterol. cBDL also leads to the downregulation of *Slc10a1* (also called Sodium taurocholate cotransporting polypeptide, *Ntcp*), which is involved in the transport of bile acids from the blood into the liver [17, 18]. This repression was hypothesized to be a protective response to prevent further accumulation of bile acids in the liver [18]. Cholestasis also modulates the expression of *Abcc2* (also called multidrug-associated protein, *Mrp2*) [17, 19] and *Abcb11* (also called bile salt efflux pump, *Bsep*) [17, 20]. Both proteins transport bile acids from the hepatocyte into the bile [21]. Other proteins, which transport bile acids from the liver into the blood, such as *Abcc3* (also called multidrug-associated protein, *Mrp3*) [17, 19] and *Abcc4* (also called multidrug-associated protein, *Mrp4*) are upregulated after cBDL [17, 22]. Consequently, cBDL leads to an accumulation of bile acids in the blood [16].

During the last years, a modification of cBDL was developed, where just the left hepatic bile duct, which drains the left liver lobe, is ligated [23–25]. This ligation type is called partial bile duct ligation (pBDL) [23, 25, 26]. Within the affected lobe pBDL causes similar to cBDL necrosis [26], inflammation [26, 27], and fibrosis [23, 26, 27]. Elevated plasma activity of liver enzymes such as ALT and ALP was observed after pBDL in transgenic mice expressing the human alpha-1 antitrypsin Z gene [25] and during the subacute phase in wildtype mice [26]. It was suggested that pBDL can be used as an animal model to study local molecular and histopathological changes after cholestasis [23, 25–27].

Apart from the translational aspect of in vivo studies, another focus must be set on the well-being of the animals used for experiments. This is a moral obligation, legally required in many countries, and also improves the quality of preclinical science [28–32]. Various non-governmental organizations, as well as scientists, pursue a complete replacement of animal experiments. However, preclinical testing in animals is still necessary before conducting clinical trials and can indeed provide important information predicting the safety and effective dosage of drugs for clinical trials [33]. For example, ursodeoxycholic acid, also known as ursodiol, is an FDA-approved drug for the treatment of primary biliary cholangitis [34, 35]. This physiologically occurring bile acid was tested for its beneficial, therapeutic, and pharmacokinetic properties in rats with induced cholestasis [36, 37]. Thereafter it was conducted successfully in clinical studies [38, 39]. Since experiments on animals can still provide important data before treating human patients, it is necessary to conduct preclinical studies and basic research while reducing the number of animals and improving their well-being. This can be achieved by

wisely choosing models, that have the appropriate pathophysiological features and mechanisms, but also have a high survival rate and cause little distress to the animals.

When evaluating the well-being of mice, various methods, such as assessing a clinical score sheet [40, 41] or reduction in body weight [42, 43], are often successfully used. Observation of behavior, like the assessment of burrowing activity, can also define the well-being of mice during an experiment [44, 45].

The primary objective of the study was to compare animal well-being between cBDL and pBDL mice. In addition, it was our aim to check if key features of cholestasis such as liver fibrosis, necrosis, inflammation and increased bilirubin, bile acids or activity of ALT, AST, ALP, and GLDH as well as altered gene expression of bile acid synthesizing enzymes and bile acid transporters can be studied in the pBDL animal model.

## Results

### Comparison of distress between cBDL and pBDL mice

The survival and distress of mice were evaluated after cBDL or pBDL. After pBDL all 14 animals survived, whereas 5 of 14 mice had to be euthanized after cBDL. Thus, the survival of mice after cBDL was significantly lower when compared to animals with pBDL (Fig 1A).

At three time points after ligation a distress score, the body weight, and the burrowing activity were compared to a pre-operative time point (pre-phase). In addition, these three methods were used to compare the distress between the two ligation methods at each time point. The distress score increased significantly after cBDL at all time points, whereas after pBDL, distress increased significantly only in the middle phase (Fig 1B). Interestingly, the distress score after pBDL was significantly lower than after cBDL at all post-operative phases.

Consistent with the evaluation of the distress score, the median body weight of cBDL animals decreases continuously during the early, middle, and late phase of cholestasis (Fig 1C). However, the median body weight of pBDL animals only dropped in the early phase after ligation and recovered during the late phase. Consequently, within the late phase the weight of cBDL animals was significantly lower than the weight of pBDL mice (Fig 1C).

Similar differences can be noticed when analyzing the burrowing activity of these mice. Mice after pBDL showed significantly reduced burrowing activity during the early phase but not during the middle and late phase (Fig 1D). Animals after cBDL burrowed significantly less in the late phase compared to the pre-operative phase. Interestingly, cBDL animals burrowed significantly less during the late phase when compared to pBDL animals (Fig 1D). These data suggest that burrowing activity indicates higher distress in cBDL animals during the late phase of cholestasis.

### Comparison of liver pathology between cBDL and pBDL

In order to evaluate liver injury, the area of necrosis was analyzed. It was significantly higher in the liver after cBDL and in the left lobes of pBDL when compared to control livers (S1A Fig). Surprisingly, the necrotic area within the ligated left lobe of the pBDL livers showed a bimodal distribution (S1 Fig). In six left liver lobes, more than 26% necrosis was observed, whereas in all other left liver lobes, less than 8% of the liver tissue was necrotic (S1 Fig). Such a bimodal distribution, as observed in S1A Fig, when analyzing pBDL mice, can occur when two distinct processes are involved in producing data. Our main hypothesis to explain this bimodal distribution was an accidental damage or ligation of the left hepatic artery, which flows alongside the left bile duct (draining the left liver lobe) [46]. To verify this hypothesis, the left bile duct of 6 animals was ligated without injuring the left hepatic artery (called verified partial bile duct ligation, v-pBDL) and 6 animals were operated with a ligation of the left bile duct plus the

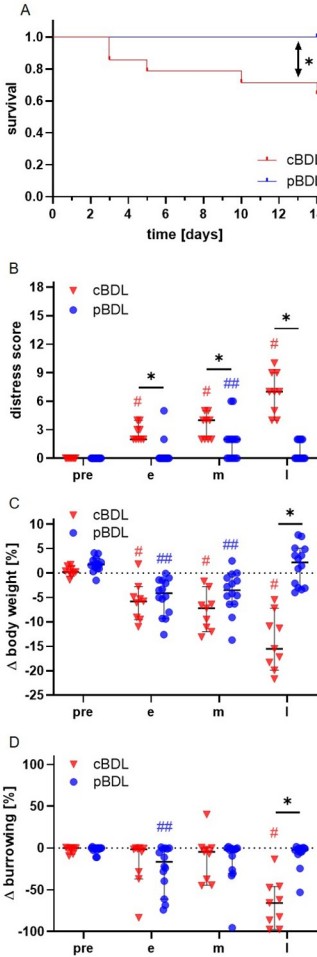

**Fig 1. Survival and distress of mice after cBDL or pBDL.** The survival rate (A) was significantly higher after pBDL than BDL; Log-rank (Mantel-Cox) test (*P < 0.05); cBDL: n = 14, pBDL: n = 14 animals. A distress score (B), body weight (C), and burrowing activity (D) were evaluated during the pre-operative phase (pre) as well as during early (e), middle (m), and late (l) phase of cholestasis. Significance in B, C and D was determined by a Two Way RM ANOVA. Sidak Test for multiple comparisons when comparing between the two ligation methods at each time point (*P < 0.05). Dunnett test for multiple comparisons when compared to pre-experimental time point within the cBDL group (#P < 0.05) and within the pBDL group (##P < 0.05). The median + 95% CI is shown; cBDL: n = 9, pBDL: n = 14 animals.

corresponding left hepatic artery (pBDL+pAL). On day 14 after v-pBDL 0.4–1.7% of the left liver lobe were necrotic, whereas after pBDL+pAL 44.5–92.3% of the left liver lobe were necrotic (S1B Fig).

## Comparison of distress between cBDL and v-pBDL mice

Consequently, we focused on comparing cBDL to v-pBDL mice. After v-pBDL all 6 animals survived, whereas 5 of 14 mice had to be euthanized after cBDL (S2A Fig). The distress score of v-pBDL, increased significantly only in the early phase, whereas it increased significantly after cBDL at all time points after ligation (S2B Fig). Consequently, the distress score after v-pBDL was significantly lower than after cBDL in the middle and late post-operative phases. The median body weight of cBDL animals decreases continuously during the early, middle,

and late phase of cholestasis (S2C Fig). However, the median body weight of pBDL animals only dropped in the early phase after ligation and started to recover in the middle and late phase. During the late phase the body weight of v-pBDL was significantly higher than the weight of cBDL animals (S2C Fig). Consistent with the distress score and body weight data animals after cBDL burrowed significantly less than v-BDL mice in the late phase (S2D Fig). These data suggest that cBDL causes more distress than v-pBDL.

## Comparison of liver pathology between cBDL and v-pBDL

We compared necrosis, collagen I deposition, and infiltration of chloroacetate esterase positive (CAE⁺) cells in the liver of healthy control animals, cBDL and v-pBDL mice. We observed no necrotic lesions in control animals, obvious big necrotic lesions in cBDL, and few small necrotic lesions in the left liver lobe of v-pBDL mice (Fig 2A–2C). The percentage of necrotic area on histological liver sections was significantly higher in cBDL mice when compared to control animals (Fig 2D).

Collagen I surrounded blood vessels and bile ducts in healthy control animals (Fig 2E), whereas cBDL and pBDL liver sections displayed additional collagen accumulation at portal areas between the liver parenchyma (Fig 2F and 2G). Both the cBDL and pBDL liver had significantly more collagen I deposition than the liver of control mice (Fig 2H).

CAE staining was performed to investigate the infiltration of neutrophil granulocytes. Control animals exhibited few CAE⁺ cells (Fig 2I), whereas cBDL (Fig 2J) and v-pBDL (Fig 2K) animals displayed pronounced infiltrations of CAE⁺ cells. Particularly, v-pBDL liver sections demonstrated significantly elevated levels of CAE⁺ cells compared to control animals (Fig 2L). In cBDL as well as in the ligated liver lobe of v-pBDL mice *TNFα* was significantly increased, while *IFNγ* was only significantly increased in the ligated liver lobe of v-pBDL animals (S3 Fig).

Liver injury was also assessed by quantifying the activity of AST, ALT, ALP as well as GLDH, and the concentration of total bile acids, total bilirubin, direct bilirubin, and indirect bilirubin on day 14 after surgical intervention. The activity of all measured enzymes was elevated in cBDL plasma samples, when compared to v-pBDL or control plasma from healthy animals (Fig 3A–3D). In pBDL plasma only GLDH activity was significantly increased when compared to control plasma (Fig 3D). Also, the concentrations of total bile acids, total bilirubin, direct bilirubin, and indirect bilirubin were elevated in cBDL mice, when compared to v-pBDL or control animals (Fig 3E, 3F and S4 Fig).

As suggested by our reviewers we assessed the expression of genes involved in bile acid synthesis (*Cyp7a1*, *Cyp8b1*, *Cyp27a*), transportation of bile acids from the blood into the liver (*Slc10a1*), transportation of bile acids from the liver into the bile (*Abcc2*, *Abcb11*) and the transportation of bile acids from the liver into the blood (*Abcc4*, *Abcc3*). The expression of *Cyp7a1*, *Cyp8b1*, *Cyp27a*, *Slc10a1*, and *Abcc2* was reduced in the liver of cBDL and the ligated liver lobe of v-pBDL mice (Fig 4A–4E), when compared to control animals. The expression of *Abcb11* and *Abcc4* is increased in cholestatic livers of cBDL and v-pBDL mice (Fig 4F and 4G), whereas the expression of *Abcc3* is increased after cBDL, but not after v-pBDL (Fig 4H).

## Distress comparison between v-pBDL and pBDL+pAL mice

To explore, if pBDL+pAL animals experience more distress than v-pBDL animals we assessed survival, distress score, body weight, and burrowing activity. All animals of the v-pBDL and pBDL+pAL group survived until the end of the experiment (Fig 5A). In addition, no significant differences in the distress score, body weight, and burrowing activity were observed between v-pBDL and pBDL+pAL mice at any of the post-operative time points (Fig 5B–5D).

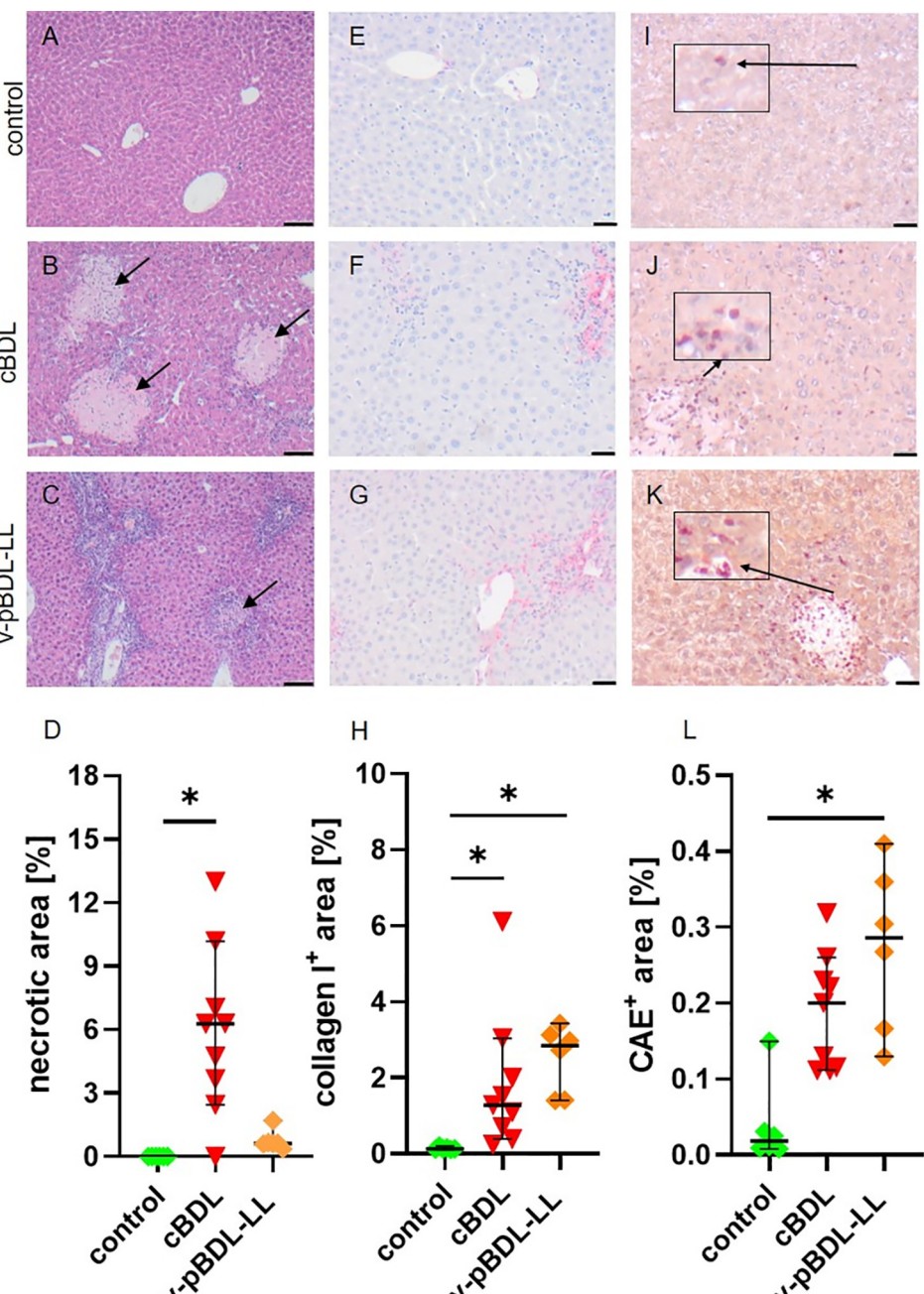

**Fig 2. Necrosis, fibrosis, and inflammation on day 14 after cBDL or v-pBDL.** Hematoxylin/eosin stained liver sections of healthy (A), cBDL (B), and v-pBDL (C) mice (black arrows indicate necrotic areas, scale bar = 50 μm) and quantification of necrosis (D). Collagen I immunohistochemistry on liver sections of healthy (E), cBDL (F), and v-pBDL (G) mice (collagen I stained in red, scale bar = 20 μm) and comparison of fibrosis between the indicated groups (H). Chloroacetate esterase staining (CAE) on liver sections of healthy (I), cBDL (J), and v-pBDL (K) mice (black arrowheads indicate CAE+ cells, scale bar = 20 μm) and quantification of CAE+ area (L). Kruskal Wallis test (ANOVA on ranks) with Dunn's correction (*P < 0.05). The median + 95% CI is shown; control: n = 6, cBDL: n = 9, v-pBDL-LL: n = 6 animals.

The distress score was in both groups the highest during the early phase after ligation (Fig 5B). The body weight was significantly decreased in both groups during the early and middle phase of cholestasis when compared to the pre-operative phase but recovered during the late phase

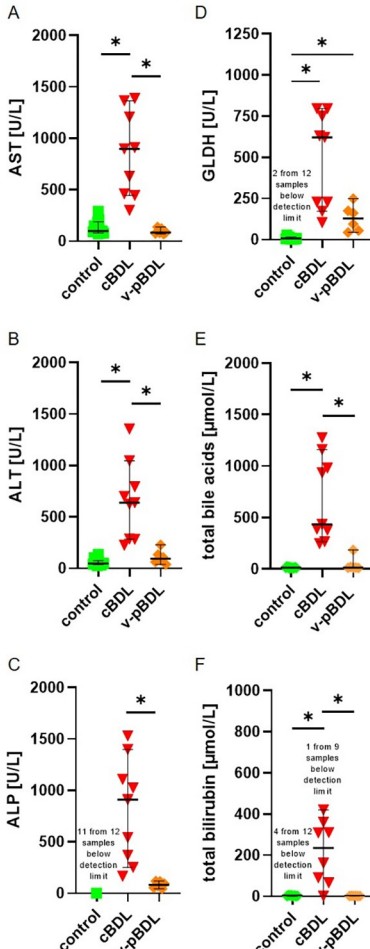

**Fig 3. Activity of liver enzymes, total bile acids, and total bilirubin concentrations in blood plasma.** AST (A), ALT (B), ALP (C), GLDH (D), total bile acids (E) and total bilirubin (F) in the blood plasma of healthy control mice and mice on day 14 after cBDL or v-pBDL are presented. Kruskal Wallis test (ANOVA on ranks) with Dunn's correction for multiple comparisons (A, B, C, D, E) or one-way ANOVA (F) with Tukey correction for multiple comparisons (*P < 0.05). The median + 95% CI is shown; control: n = 12, cBDL: n = 9, v-pBDL: n = 6 animals.

(Fig 5C). Burrowing activity was only decreased significantly in the pBDL+pAL group during the early phase of cholestasis when compared to the pre-operative phase (Fig 5D). Thus, no major differences between pBDL and pBDL+pAL animals were observed when analyzing distress using distinct methods.

## Liver pathology in v-pBDL and pBDL+pAL mice

When assessing the liver, already macroscopic observations revealed major differences between the left liver lobes of v-BDL and pBDL+pAL mice (S5 Fig). Histological sections confirmed focal necrosis after v-pBDL (Fig 6A) and large areas of necrosis after pBDL+pAL (Fig 6B). Quantification of the necrotic area revealed that pBDL+pAL induced a lot of necrosis (median: 67.9%, 95% confidence interval: 44.5–92.3%), whereas after v-pBDL little necrosis (median: 0.6%; 95% confidence interval: 0.4–1.7%) was observed (Fig 6C). No significant differences of necrosis were noticed when analyzing the non-ligated middle liver lobe of v-pBDL or pBDL+pAL mice (S6A–S6C Fig).

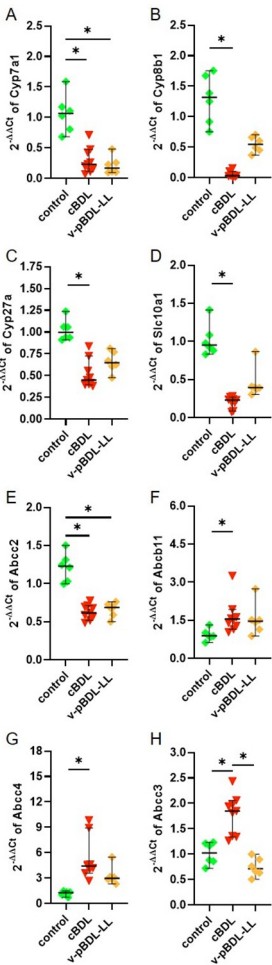

**Fig 4. Expression of genes involved in bile acid synthesis and bile acid transport in the ligated liver lobes of cBDL and v-pBDL mice.** Relative expression ($2^{(-\Delta\Delta CT)}$) of *Cyp7a1* (A), *Cyp8b1* (B), *Cyp27a* (C), *Slc10a1* (D), *Abcc2* (E), *Abcb11* (F), *Abcc4* (G) and *Abcc3* (H). One-way ANOVA with Tukey correction (A, E, H) or Kruskal Wallis test with Dunn's correction (B, C, D, F, G) was done (*$P < 0.05$). The median + 95% CI is shown; control: n = 6, cBDL: n = 9, v-pBDL: n = 6 animals.

Subsequently, fibrosis was analyzed by collagen I immunohistochemistry in the liver tissue. Interestingly, v-pBDL liver sections displayed collagen accumulation at and in between portal areas (Fig 6D). However, collagen depositions in liver sections after pBDL+pAL were found mainly at the edge of necrotic tissue (Fig 6E). In the ligated liver lobe of both v-pBDL and pBDL+pAL mice more collagen was detected than in control mice (Fig 6F). In contrast, no major increase in the collagen I$^+$ area was noticed when comparing the non-ligated middle liver lobe of v-pBDL or pBDL+pAL mice to the liver of control animals (S6D–S6F Fig).

Similar to collagen deposition, CAE staining revealed an increase in CAE$^+$ cells in both v-pBDL and pBDL+pAL liver sections compared to controls (Fig 6G and 6H). Notably, v-pBDL liver sections exhibited a significantly higher number of CAE$^+$ cells when compared to control samples (Fig 6I). In contrast, no major increase in CAE$^+$ cells was noticed when comparing the non-ligated middle liver lobe of v-pBDL or pBDL+pAL mice to the liver of control animals (S6G–S6I Fig). In the ligated liver lobe of v-pBDL as well as pBDL+pAL mice, *TNFα* and *IFNγ*

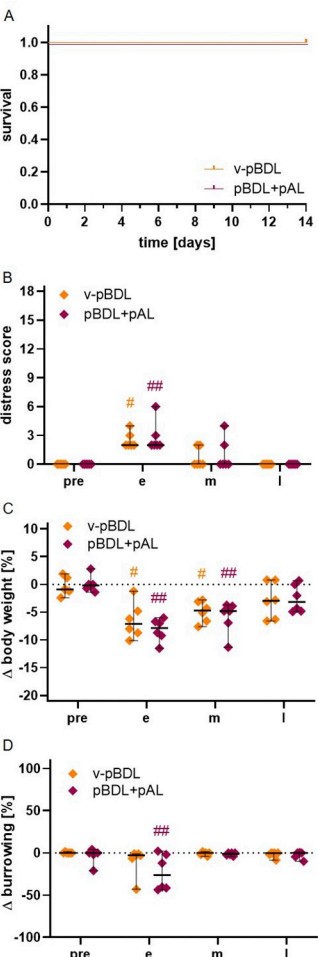

**Fig 5. Survival and distress of mice after v-pBDL or pBDL+pAL.** After the indicated surgical interventions, the survival rate (A), the distress score (B), body weight (C), and burrowing activity (D) were evaluated at the indicated days or during the pre-operative phase (pre) as well as during early (e), middle (m) and late (l) phase of cholestasis. Significance was determined by Log-rank (Mantel-Cox) test (A) or a Two Way RM ANOVA (B-D) followed by Sidak Test for multiple comparisons when comparing between the two ligation methods at each time point (*P < 0.05) or a Dunnett test for multiple comparisons when compared to pre-experimental time point within the cBDL group (#P < 0.05) and within the pBDL group (##P <0.05). The median + 95% CI is shown; v-pBDL: n = 6, pBDL+pAL: n = 6 animals.

were increased (S7 Fig). In the unligated liver lobes of v-pBDL as well pBDL+pAL mice, only a minor non-significant increase in *TNFα* and *IFNγ* was observed (S8 Fig).

These data demonstrate that pBDL, without injuring the left hepatic artery (v-pBDL), as well as pBDL+pAL induces liver inflammation. Focal necrosis and fibrosis with fibrotic connections between portal areas are observed in the ligated liver lobe of v-pBDL mice, whereas pBDL+pAL induces large areas of necrosis and fibrosis, often surrounding the necrotic tissue.

The activity of AST, ALT, ALP as well as GLDH and the concentration of total bile acids and bilirubin were also assessed. No major increase in the activity of AST, ALT, and ALP was observed, whereas GLDH was significantly increased in v-pBDL or pBDL+ pAL mice when compared to control animals (Fig 7A–7D). Total bile acids and total bilirubin were also not increased in v-pBDL or pBDL+ pAL mice when compared to control animals (Fig 7E and 7F).

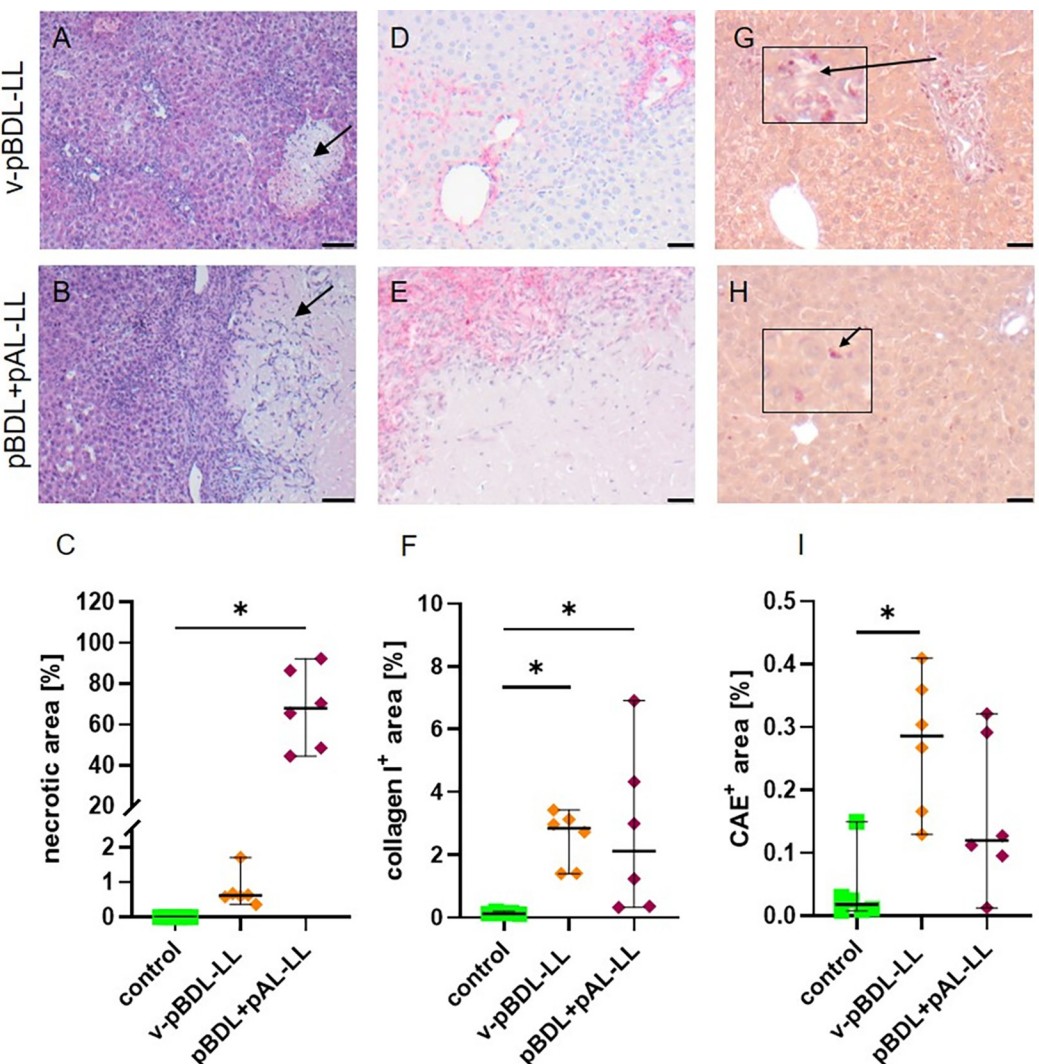

**Fig 6. Evaluation of liver necrosis, fibrosis, and inflammation in ligated liver lobes.** Hematoxylin/eosin stained liver sections of v-pBDL (A) and pBDL+pAL (B) mice (black arrows indicate necrotic areas (scale bar = 50 μm) and quantification of necrosis (C). Collagen I immunohistochemistry on liver sections of v-pBDL (D) and pBDL+pAL (E) mice (collagen I is stained in red, scale bar = 20 μm) and comparison of fibrosis between the indicated groups (F). Chloroacetate esterase staining (CAE) on liver sections of v-pBDL (G) and pBDL+pAL (H) mice (black arrowheads indicate CAE$^+$ cells, scale bar = 20 μm) and quantification of CAE$^+$ area (I). Kruskal Wallis test (ANOVA on ranks) with Dunn's correction; The median + 95% CI is shown; control: n = 6, cBDL: n = 9, v-pBDL-LL: n = 6 animals.

Between v-pBDL or pBDL+ pAL mice, no significant differences in AST, ALT, ALP, GLDH, total bile acids, total bilirubin, direct bilirubin, or indirect bilirubin were noticed (Fig 7A–7F and S9 Fig).

We assessed the expression of genes involved in bile acid synthesis and transportation of bile acids in the ligated liver lobes of v-pBDL and pBDL+pAL mice. The expression of *Cyp7a1*, *Cyp8b1*, *Cyp27a*, *Slc10a1*, and *Abcc2* was significantly reduced in these cholestatic mice (Fig 8A–8E) when compared to control animals. The expression of *Abcb11* is not significantly increased (Fig 8F), whereas *Abcc4* is significantly increased in the ligated liver lobes of v-pBDL and pBDL+pAL mice (Fig 8G). The expression of *Abcc3* was non-significantly reduced in v-pBDL or pBDL+pAL animals (Fig 8H). We also assessed the expression of these genes in the

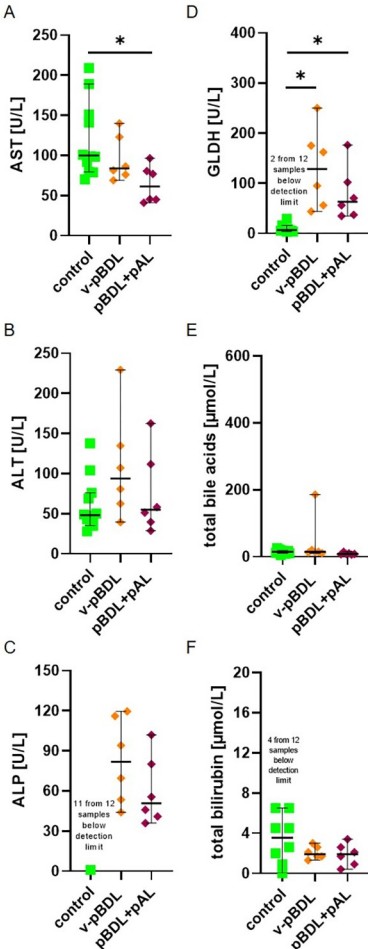

**Fig 7. Activity of liver enzymes, total bile acid, and bilirubin concentrations in blood plasma of v-pBDL and pBDL+pAL mice.** AST (A), ALT (B), ALP (C) and GLDH (D), total bile acids (E), and total bilirubin (F) of healthy control animals and mice on day 14 after v-pBDL or pBDL+pAL. Kruskal Wallis test (ANOVA on ranks) with Dunn's correction (A-E) or one-way ANOVA with Tukey correction (F) for multiple comparisons (*P < 0.05). The median + 95% CI is shown; control: n = 12, v-pBDL: n = 6, pBDL+pAL: n = 6 animals.

unligated liver lobes of v-pBDL and pBDL+pAL mice (Fig 9). The expression of *Cyp7a1*, *Cyp8b1*, and *Slc10a1* but not *Cyp27a* was significantly reduced in these cholestatic animals when compared to healthy mice (Fig 9A–9D). *Abcc2* expression was also reduced (Fig 9E), whereas *Abcb11* expression was increased (Fig 9F). Little changes were observed when assessing *Abcc4* expression (Fig 9G), whereas the expression of *Abcc3* was reduced in pBDL+pAL animals (Fig 9H).

## Discussion

The results of this study demonstrate that mice after cBDL experience more distress and a higher mortality than mice after pBDL (Fig 1 and S2 Fig). However, no differences in survival or distress were observed when comparing v-pBDL to pBDL+pAL (Fig 5).

More distress after cBDL compared to pBDL might be caused by impaired detoxification, due to interrupted ammonium metabolism [47] or bilirubin elimination [2]. In addition,

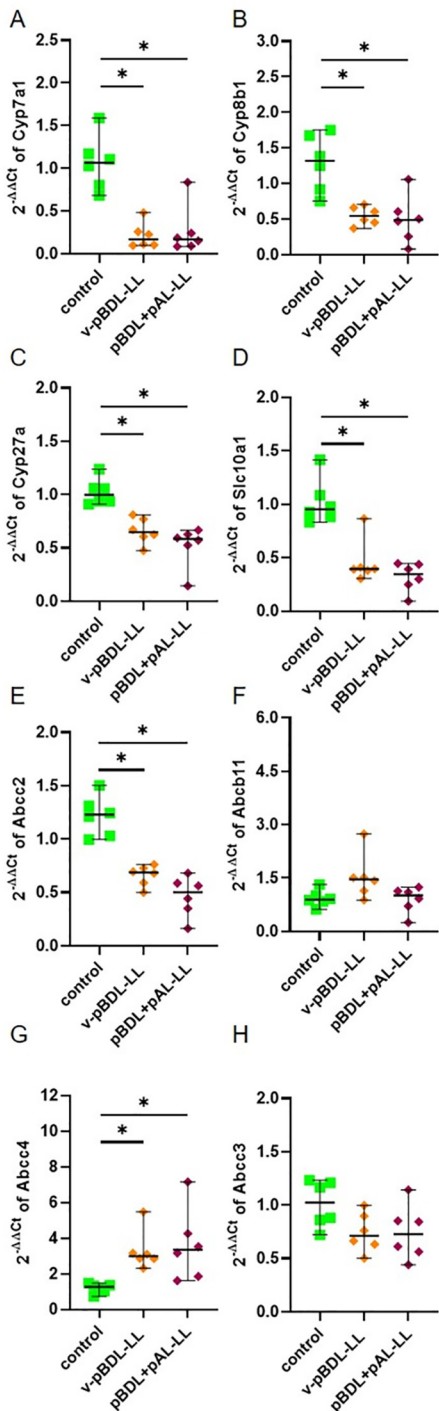

**Fig 8. Expression of genes involved in bile acid synthesis and bile acid transport in ligated liver lobes.** Relative expression ($2^{(-\Delta\Delta CT)}$) of *Cyp7a1* (A), *Cyp8b1* (B), *Cyp27a* (C), *Slc10a1* (D), *Abcc2* (E), *Abcb11* (F), *Abcc4* (G), and *Abcc3* (H) in the left liver lobe (-LL) in v-pBDL and pBDL+pAL mice. One-way ANOVA with Tukey correction (B, E, F, H) or Kruskal Wallis test with Dunn's correction (A, C, D, G) for multiple comparisons (*$P < 0.05$). The median + 95% CI is shown; control: n = 6, v-pBDL-LL: n = 6, pBDL+pAL: n = 6 animals.

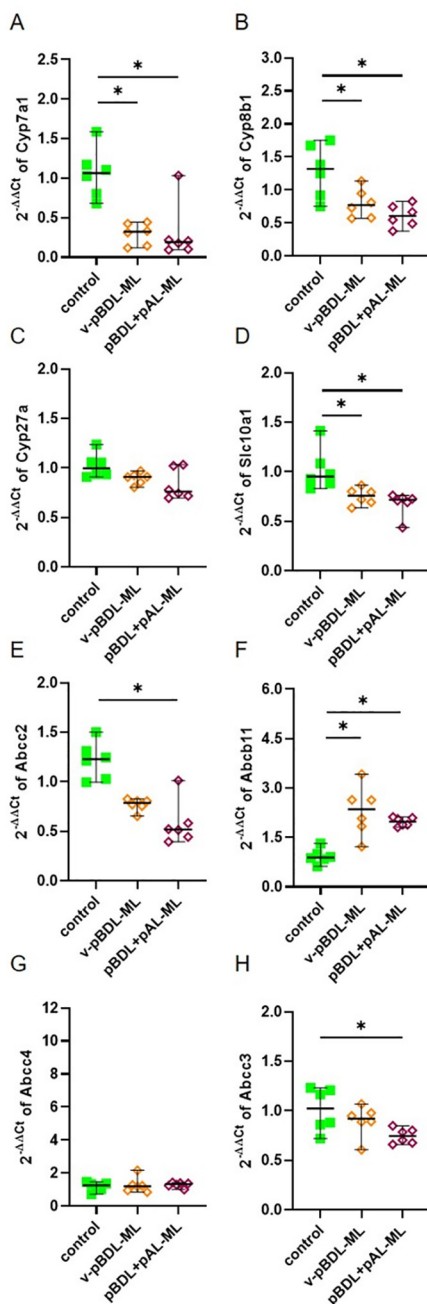

**Fig 9. Expression of genes involved in bile acid synthesis and bile acid transport in unligated liver lobes.** Relative expression ($2^{(-\Delta\Delta CT)}$) of *Cyp7a1* (A), *Cyp8b1* (B), *Cyp27a* (C), *Slc10a1* (D), *Abcc2* (E), *Abcb11* (F), *Abcc4* (G), and *Abcc3* (H) in the median liver lobe (-ML) in v-pBDL and pBDL+pAL mice. One-way ANOVA with Tukey correction (B, F, G, H) or Kruskal Wallis test with Dunn's correction (A, C, D, E) for multiple comparisons (*P < 0.05). The median + 95% CI is shown; control: n = 6, v-pBDL-LL: n = 6, pBDL+pAL: n = 6 animals.

cBDL completely prevents bile from entering the intestine, leading to a reduction in fat absorption [48], which might contribute to body weight reduction [47]. In addition, the cBDL animals also had a significantly higher mortality than pBDL mice (Fig 1 and S2 Fig). A high

**Table 1. Pathological features of cholestasis, primary biliary cholangitis, primary sclerosing cholangitis, biliary atresia, and acute liver failure.**

| Pathological feature | cholestasis | primary biliary cholangitis | primary sclerosing cholangitis | biliary atresia | acute liver failure |
|---|---|---|---|---|---|
| Necrosis | Focal necrosis: [58] | Focal necrosis: [59] | Focal necrosis: [60] | Focal necrosis: [61] | Confluent necrosis: [68] |
| Fibrosis | [62] | [59, 63] | [60, 64] | [65] | [68] |
| Local Inflammation | [62, 66] | [59, 63] | [60] | [65] | [68] |
| Elevated activity of liver enzymes in the blood | [58, 67] | [59] | [60] | [65] | [85] |

mortality after cBDL has also been reported in several other studies [49–51] Possible reasons have been widely debated. For example, it was suggested that the transection of the bile duct carries the risk of bile acid leakage, which can lead to peritonitis or sepsis [52, 53]. Other studies mentioned that the development of trauma to other organs such as the pancreas during the performance of surgery may result in sepsis [54]. In addition, it has been described that cBDL leads to increased intestinal permeability for bacteria, which might lead to complications [55].

When comparing v-pBDL to pBDL+pAL animals, no statistical difference in survival rate, the distress score, the body weight, or burrowing activity was observed (Fig 5), although pBDL +pAL animals had much more liver necrosis (Fig 6A–6C). Thus we conclude that increased tissue necrosis in the left liver lobe does not lead to more distress or a higher mortality. This conclusion is supported by a study by Chen et al. [56]. Chen et al. assessed liver injury and necrosis after feeding different dosages of mycotoxins. The higher the dosages the more lesions within liver tissue were scored. However, they did not observe a correlation between increased lesions and body weight reduction [56].

Besides issues of animal welfare, it is also very important to assess how suitable an animal model is for studying a specific disease or pathophysiological mechanisms. For this purpose, it is important to compare the pathophysiological features between a human disease and specific animal models [57]. During obstructive cholestasis, primary biliary cholangitis, primary sclerosing cholangitis, and biliary atresia, pathological features such as focal necrosis, fibrosis, inflammation, and elevated activity of liver enzymes in the blood can be observed (Table 1) [58–67]. During acute liver failure fibrosis, inflammation and elevated activity of liver enzymes in the blood can also be noticed. However, acute liver failure can lead to confluent necrosis [68].

The cBDL animal model allows to study focal necrosis, fibrosis, and inflammation in the liver, as well as elevated activity of liver enzymes, total bile acids, and bilirubin concentration in the blood (Table 2) [2–4, 69]. This animal model is, therefore, used to study the above mentioned diseases [1–3]. After pBDL, the same local pathological features can be observed in the

**Table 2. Occurrence of pathological features in animal models.**

| Pathological feature | cBDL | pBDL | pBDL+pAL |
|---|---|---|---|
| Necrosis | + (focal) Fig 2B, [4] | + (focal) Fig 2C, [26] | + (confluent) Fig 6B |
| Fibrosis | + Fig 2F, [2–4] | + Fig 2G, [23, 26] | + Fig 6E |
| Local Inflammation | + Fig 2J, [3, 69] | + Fig 2K, [26] | Fig 6H |
| Elevated activity of liver enzymes in the blood | + Fig 3, [2–4] | (+) Fig 3 [a], Fig 7 [a], [23, 26] | (+) Fig 7[a] |

[a]increased GLDH plasma activity

liver (Table 2) [23, 25]. However, in contrast to cBDL, no major increase in total bile acids and bilirubin concentration or the activity of liver enzymes such as AST, ALT, and ALP was noticed in the blood on day 14 after pBDL (Fig 3). Thus, the pBDL animal model can be used to study local features of the mentioned diseases, but is less suited to study systemic features of cholestasis compared to the cBDL animal model.

After pBDL+pAL pathological features such as necrosis, fibrosis, inflammation, and a significant elevation of GLDH plasma activity can be observed (Fig 6, S7 Fig and Table 2). However, the main difference to the other two studied animal models is the development of large areas of necrosis within the ligated liver lobe (Fig 6B and 6C). Interestingly loss of arterial blood supply alone has been reported not to lead to major necrosis in the liver [70, 71]. Thus, we assume that the confluent necrosis we observed in pBDL+pAL mice is caused by interrupting the arterial blood supply and at the same time inducing cholestasis. This conclusion is also supported by a study, which described the effect of hepatic artery ligation in the presence or absence of cholestasis [72]. Possibly, the loss of arterial blood supply to the vascular peribiliary plexus causes dysfunction of the bile duct epithelial cells leading to their cell death, which then results in unhindered bile leakage and more severe necrosis of liver tissue [73]. This could lead to fast necrosis of the left bile duct and the surrounding liver tissue. Consequently, the pBDL +pAL animal model could be used for studying local features of acute liver failure (Table 1), which is characterized by confluent necrosis [68, 74]. In addition, pBDL+pAL might mimic biliary injuries associated with vascular injuries, which is observed in view patients after cholecystectomy or hemobilia followed by hepatic artery embolization [75, 76].

Pathophysiological processes after cholestasis also regulate the expression of genes involved in bile acid synthesis or the transportation of bile acids in form of an adaptive response. Indeed, *Cyp7a1*, *Cyp8b1*, *Cyp27a*, *Slc10a1*, Abcc2, *Abcb11*, and *Abcc4* were similarly regulated in both cBDL as well as v-pBDL mice, when compared to controls (Fig 4). For example, *Cyp7a1* expression was significantly repressed in the ligated liver lobes after v-pBDL or cBDL (Fig 4A). Possibly, this is caused via well-described farnesoid X receptor (FXR) dependent mechanisms within the liver [77, 78]: Bile acids bind to FXR in hepatocytes, which represses *Cyp7a1* expression by mechanisms partially dependent on the small heterodimer partner SHP [77, 79]. A second well-described mechanism by which *Cyp7a1* expression can be repressed depends on an enterohepatic signal. Bile acids bind to FXR in the intestine and induce the expression of fibroblast growth factor 15 (*FGF15*) / fibroblast growth factor 19 (*FGF19*) [77, 79], which then cause the suppression of *Cyp7a1* expression in the liver. We observed a similar strong downregulation of *Cyp7a1* in v-pBDL mice (after a surgical intervention, that allows drainage of bile from the other liver lobes to the intestine) when compared to cBDL animals (after intervention that should lead to a strong reduction of bile acids in the intestine and therefore rather to a de-repression than a repression of *Cyp7a1* expression). Thus, we argue, that such an enterohepatic signal does not have a major function in *Cyp7a1* repression observed in v-pBDL or cBDL mice. Interestingly, we also observed repression of *Cyp7a1* in the unligated middle liver lobe of v-pBDL mice (Fig 9). Probably this observation cannot be explained by local mechanisms or enterohepatic signals described above. However, a third mechanism was suggested of how *Cyp7a1* expression can be repressed. It has been proposed, that bile acids induce the expression of inflammatory cytokines such as *TNFα*, which then inhibits *Cyp7a1* expression [77, 80, 81]. This pathway could potentially also regulate gene expression in unligated liver lobes since *TNFα* can also have systemic effects [82]. Our data demonstrating a strong upregulation of *TNFα* in pBDL mice is consistent with such a hypothesis (S3 Fig).

However, the results presented in this publication should not be over-interpreted, since only one late time point (day 14 after ligation) was examined when the tissue was already

partially necrotic. The concept that exploration of gene expression during cholestasis can be problematic is also supported by contradictory results observed in the literature. For example, one publication demonstrates that cBDL significantly reduces *Abcc2* (*Mrp2*) expression on day 14 after cBDL [19]. This is consistent with the data presented in Fig 4E. However, another publication demonstrates a significant induction of *Abcc2* (*Mrp2*) on day 14 after BDL [17]. In a very similar manner, *Abcb11* (*Bsep*) expression was published to be significantly induced on day 7 and non-significantly induced on day 14 after cBDL [17]. This result is consistent with the data presented in Fig 4D. However, other studies published no induction of this gene at day 14 of cholestasis [19, 83]. Possibly, such results are influenced by various variables such as sex, the mouse strain, or different euthanasia criteria applied during distinct projects. Thus, we suggest that additional studies need to confirm the observed gene regulations, (e.g. evaluation of expression at various time points, using both qPCR and Western Blots) and refrain from discussing all other gene expression results.

In this study, we demonstrated that pBDL is more beneficial than cBDL for the well-being of mice. However, it is a disadvantage of the pBDL animal model that it is technically more difficult, and that there is the risk of damaging the left hepatic artery, which runs very close to the left bile duct [46, 84]. Blocking this artery has a dramatic influence on the liver. It causes a lot more necrosis (Fig 6C). However, the pBDL animal model also offers advantages when compared to cBDL. For example, pBDL is more beneficial for the well-being of the animals and leads to a significantly higher survival rate. Thus, even vulnerable genetically altered mice can be used to study certain human diseases [25, 26]. In addition, our data (Fig 1A) suggest that a lower number of animals will suffice when evaluating cholestasis using pBDL instead of cBDL, since fewer animals need to be euthanized when using this animal model. For future studies, it would be desirable to increase the use of pBDL as a replacement for the cBDL animal model. However, the similarity of an animal model to a certain disease and the technical skill of the surgeon must also be considered, when deciding between these two animal models.

## Material and methods

### Animals

BALB/cANCrl mice were originally purchased from Charles River and bred in the facility of the University Medical Center in Rostock. The health of the animal stock is routinely checked (Helicobacter sp., Rodentibacter pneumotropicus, murine Norovirus, and rat Theilovirus were detected within the last two years in the animal facility; unhealthy animals were not used for any experiments). Only male mice (age: 79/72-90 median/interquartile range in days) were used for this experiment since female mice were needed for expanding the strain. They were allowed to acclimatize (more than two days before practicing burrowing and more than 12 days before surgical intervention). All mice were single-housed in Eurostandard Type III clear plastic cages, under controlled temperature (21 ± 2°C), humidity (60 ± 20%) and a light-dark cycle of 12/12 h (dawn: 6:30–7:00). Food pellets (V1534.000, 10 mm, ssniff Spezialdiäten GmbH, Soest, Germany) and tap water were freely available. We also put autoclaved bedding (Bedding Espe Max 3–5 mm granulate, H 0234–500, Abedd, Vienna, Austria), shredded tissue paper (PZN03058052, FSMED Verbandmittel GmbH, Frankenberg, Germany), one paper tunnel (75 × 38 mm, H 0528–151, ssniff Spezialdiäten GmbH, Soest, Germany) and a wooden enrichment tool (Espe size S, 40 × 16 × 10 mm, H0234.NSG, Abedd, Vienna, Austria) into each cage. The sample sizes were calculated and the research question was defined when applying for the permission to do these experiments. The primary outcome measure was burrowing activity for assessing animal welfare. All experiments were approved by Landesamt für Landwirtschaft, Lebensmittelsicherheit und Fischerei Mecklenburg-Vorpommern and were

conducted in accordance with the European directive 2010/63/EU as well as the national law of Germany. The authors complied with the ARRIVE guidelines.

## Induction of cholestasis

The mice were anesthetized by 1.2–2.5 vol. % isoflurane (CP-pharma, Burgdorf, Germany) and kept on a warming plate at 37˚C during the operation. In order to relieve the pain, the recommended dose of 5 mg/kg Carprofen (Pfizer GmbH, Berlin, Germany) was injected subcutaneously once before laparotomy. An eye ointment was applied for protecting the eyes from drying out and an approximate 2 cm midline abdominal incision was made. Sutures and retractors were used to secure the operator's field of view. Then all procedures were performed under a Leica M851 manual surgical microscope (Leica, Wetzlar, Germany) at a magnification of 10–16 folds. In order to develop the cBDL model, the common bile duct was carefully separated from the proper hepatic artery and the portal vein with a micro-dissecting forceps. The bile duct was then ligated by three ligations (5–0, Polyester-S, Catgut GmbH, Markneukirchen, Germany) and intersected (S10A–S10C Fig). No organ damage, rupture of the hepatic artery/portal vein, or active bleedings were observed. The peritoneum and skin were then closed by 6–0 and 4–0 polypropylene sutures, respectively. The entire surgical intervention took 25–40 minutes. Necrosis of the small gut, pancreas, or the mesenterium were macroscopically not observed after euthanizing the animals. To develop the pBDL model, the left hepatic bile duct, which drains the left lobe, was separated from the accompanying artery and portal vein. It was then ligated with 10–0 monofil polyamide (S10D Fig). For the pBDL + pAL model, the left hepatic bile duct and the accompanying artery were ligated with 10–0 monofil polyamide. The peritoneum and the skin were closed by 6–0 and 4–0 polypropylene suture, respectively. Each mouse was allowed to recover from anesthesia in a single cage in front of a red warming lamp. The entire surgical intervention took 40–60 minutes. After both surgical interventions 0.25 ml metamizol (500 mg/ml, Ratiopharm GmbH, Ulm, Germany) was added to the drinking water (100 ml, drinking water was changed daily). In order to minimize age as cofounder effect, the animals were allocated in a non-random manner to the groups, because we needed to match the age between each group. Four pBDL animals were excluded from data evaluation (although these animals survived until day 14) due to problems during surgical intervention (failed ligation: n = 1, necrosis in median liver lobe: n = 2) or random development of a subcutaneous abscess (n = 1). Mice were euthanized by blood removal plus cervical dislocation after anesthesia using 2.5–5 vol. % isoflurane on day 14 or when one of the humane endpoint criteria was met according to the used distress score sheet.

## Assessment of well-being

**Burrowing activity.**   The burrowing behavior was used for evaluating the well-being of mice according to Deacon [45]. A tube (15 cm × 6.5 cm) filled with 200 g pellets (V1534.000, 10 mm, ssniff Spezialdiäten GmbH, Soest, Germany) was placed in the cage 2–3 hours before the dark phase. After 17 ± 2 hours, the pellets remaining in the tube were weighted and the burrowed pellets were calculated. At the beginning of the experiment, the mice were allowed to practice burrowing activity two times. After this learning period, burrowing activity was evaluated before surgery (pre) as a baseline and at different phases of disease progression: early (day 1), middle (day 4), and late (day 13). For details see S11 Fig.

**Distress score and body weight.**   Based on a scoring table published in previous studies [40, 41], we have evaluated the health status of mice (e.g. appearance, spontaneous and flight behavior) daily with the help of a distress score. In Figs 2 and 5, the distress score is presented before surgery (pre) and during early (day 1), middle (day 4) as well as late (day 13) phase of

disease progression (S11 Fig). This score was also used to define humane endpoints. The animals were euthanized when one of the following criteria was observed: More than 25% body weight loss, abnormal respiratory sounds, the animal is cold, self-mutilation, distinct apathy or hyperkinetic, ascites, or reaching a total score of more than 15. The body weight was also evaluated before surgery (pre), and during early (day 2), middle (day 5) as well as late (day 14) phase of disease progression (S11 Fig). Since reduced food uptake in response to a certain distress level takes one day to lead to a reduction in body weight, the body weight was measured one day after we measured the distress score.

## Assessment of liver damage

**Analysis of blood plasma.** Blood was taken on day 14 by retroorbital puncture of anesthetized mice, centrifuged and the plasma was stored at -80°C. In order to evaluate liver damage, the ALT, AST, GLDH, ALP, total, and direct bilirubin activity were spectrophotometrically assessed using the Cobas c111 analyzer (Roche GmbH, Mannheim, Germany). Indirect bilirubin was calculated (total bilirubin—direct bilirubin). Bile acid concentration was determined by using the mouse total bile acid kit (Chrystal Chem, Elk Grove Village, USA).

**Assessment of liver lobes.** On day 14, tissue samples of the left and median liver lobes were either frozen or fixed in 4% buffered formalin. To evaluate inflammation, characterized by neutrophil granulocyte infiltration, CAE staining was conducted. For this procedure, 10 mg of fast garnet (Merck KGaA, Darmstadt, Germany), dissolved in 100 ml of phosphate-buffered saline, was combined with 16 mg of naphthol AS-D-chloroacetate (Merck KGaA, Darmstadt, Germany) dissolved in 2 ml of DMSO, and then filtered. The histologic sections were incubated in this solution, rinsed, washed, and stained with hematoxylin (Merck KGaA, Darmstadt, Germany). To assess necrotic areas, tissue sections were stained with hematoxylin-eosin. Fibrosis was judged after immunohistochemistry using a rabbit anti-collagen I antibody (Abcam AG, Cambridge, United Kingdom, code: ab270993, 500x dilution) and a phosphatase coupled goat anti-rabbit antibody (Abcam AG, code: ab97048, 200x dilution). For each liver lobe, 10–15 photos were taken with a BX51 microscope and a SC50 camera (Olympus, Hamburg, Germany). The percentage of the necrotic or fibrotic area was calculated with the help of Image J 1.52a software (National Institutes of Health, Bethesda, USA). The scientist taking the photos and performing the automated evaluation with Image J was blinded and therefore not aware of the group allocation.

For quantitative real-time polymerase chain reaction (TaqMan RT-qPCR), RNA was extracted from the liver lobe of each mouse using QIAzol lysis reagent and the RNeasy Mini Kit (both from Qiagen, Hilden, Germany). cDNA was synthesized using the High-Capacity cDNA Reverse Transcription Kit (Applied Biosystems, Waltham, USA). The calibrator consisted of an RNA pool isolated from 6 control BALB/cANCrl mice. After the synthesis of cDNA, Taqman RT-qPCR was performed. TaqMan gene expression assays from Thermo Fisher Scientific (Waltham, MA, USA) quantified *Abcc2* (Mm00496899_m1), *Abcc3* (Mm00551550_m1), *Abcc4* (Mm01226372_m1), *Abcb11* (Mm00445168_m1), *Slc10a1* (Mm00441421_m1), *Cyp7a1* (Mm00484150_m1), *Cyp8b1* (Mm00501637_s1), *Cyp27a1* (Mm00470430_m1), *TNFα* (Mm00443258_m1), and *IFNγ* (Mm99999071_m1), with *GAPDH* (Mm99999915_g1) serving as the reference gene. Ct values were calculated using the QuantStudio software (Applied Biosystems, Waltham, USA). ΔCt (Ct$_{gene\ of\ interest}$−Ct$_{reference\ gene}$ and ΔΔCt (ΔCt - ΔCt$_{calibrator}$) and graphed as fold changes in expression ($2^{(-\Delta\Delta Ct)}$).

**Data presentation and statistical analysis.** GraphPad Prism 8.4.3 (GraphPad Software, San Diego, USA) was utilized to create the graphs and perform the statistical analysis. The survival time of cBDL, pBDL, and pBDL+pAL mice are presented as Kaplan-Meier curves, and

the P-value was determined by log rank (Mantel Cox) test. All other data are presented in the form of point plots, indicating median ± 95% confidence interval. The Shapiro-Wilk test was applied for the assessment of normality. When normality was given, a one-way ANOVA with Tukey correction was used to determine the P-values. In case normality was not given, the Kruskal-Wallis test with Dunn's correction for comparing more than two groups or the Mann-Whitney test for comparing two groups was used. When identical animals were evaluated at different time points, a two-way RM ANOVA plus Sidak's test or Dunnett's for correcting multiple comparisons was used.

## Supporting information

**S1 Fig. Quantification of liver damage.** The percentage of necrotic areas (A) was assessed in healthy mice (control) after cBDL and within the ligated left liver lobes of pBDL mice (pBDL-LL). Necrosis was also evaluated (B) after ligating the left bile duct without injuring the left hepatic artery (called verified partial bile duct ligation, v-pBDL) or after ligating the left bile duct plus the left hepatic artery (pBDL+pAL-LL). Kruskal Wallis test (ANOVA on ranks) with Dunn's correction (A) and Mann-Whitney test (B). *P < 0.05 is considered to be statistically significant. The median + 95% CI is shown; control: n = 6, cBDL: n = 9, pBDL-LL: n = 14. (DOCX)

**S2 Fig. Survival and distress of mice after cBDL or v-pBDL.** After cBDL or v-pBDL the survival rate (A), the distress score (B), body weight (C), and burrowing activity (D) were evaluated during the pre-operative phase (pre) as well as during early (e), middle (m) and late (l) phase of cholestasis. In B, C and D the significance was determined by a Two-Way RM ANOVA. Sidak Test for multiple comparisons when comparing between the two ligation methods at each time point (*P < 0.05). Dunnett test for multiple comparisons when compared to pre-experimental time point within the cBDL group (#P < 0.05) and within the v-pBDL group (##P < 0.05). The median + 95% CI is shown; cBDL: n = 9, v-pBDL: n = 6 animals. (DOCX)

**S3 Fig. Cytokine expression in cBDL and v-pBDL mice.** Relative expression ($2^{(-\Delta\Delta CT)}$) of *TNFα* (A) and *IFNγ* (B) in the liver of healthy mice (control), after cBDL and within the ligated left liver lobe of v-pBDL mice (v-pBDL-LL). Kruskal Wallis test (ANOVA on ranks) with Dunn's correction for multiple comparisons (*P < 0.05). The median + 95% CI is shown; control: n = 6, cBDL: n = 9, v-pBDL: n = 6 animals. (DOCX)

**S4 Fig. Direct and indirect bilirubin.** Direct (A) and indirect bilirubin (B) in blood plasma of healthy mice (control), after cBDL and v-pBDL. Since in many samples bilirubin was below the detection limit, no statistical evaluation was done. The median + 95% CI is shown; control: n = 12, cBDL: n = 9, v-pBDL: n = 6 animals. (DOCX)

**S5 Fig. Liver necrosis.** Macroscopic images of liver show necrosis (black arrows) on day 14 after v-pBDL (A) or pBDL+pAL (B). (DOCX)

**S6 Fig. Evaluation of necrosis, fibrosis, and inflammation in unligated liver lobes.** Histological sections of the median liver lobes of v-pBDL (v-pBDL-ML) and pBDL+pAL (pBDL+pAL-ML) mice were analyzed. Hämatoxylin/eosin stained sections (A, B), the percentage of necrotic area (C), representative images after collagen I immunohistochemistry (D, E), the percentage of collagen I$^+$ area (F), chloroacetate esterase stained sections (G, H) and its

quantification (I) are presented. No significant differences were observed using Kruskal Wallis test (ANOVA on ranks) with Dunn's correction. The median + 95% CI is shown; control: n = 6, v-pBDL-ML: n = 6, pBDL+pAL-ML: n = 6 animals.
(DOCX)

**S7 Fig. Expression of cytokines in ligated liver lobes.** Relative expression ($2^{(-\Delta\Delta CT)}$) of TNFα (A) and IFNγ (B) in the liver of healthy mice (control), and in the ligated left liver lobes of v-pBDL (v-pBDL-LL) and pBDL+pAL (pBDL+pAL-LL) mice. Kruskal Wallis test (ANOVA on ranks) with Dunn's correction (A) or one-way ANOVA (B) with Tukey correction for multiple comparisons (*$P < 0.05$). The median + 95% CI is shown; control: n = 6, v-pBDL: n = 6, pBDL +pAL: n = 6 animals.
(DOCX)

**S8 Fig. Expression of cytokines in unligated liver lobes.** Relative expression ($2^{(-\Delta\Delta CT)}$) of TNFα (A) and IFNγ (B) in the liver of healthy mice (control), and in the unligated medium liver lobes of v-pBDL (v-pBDL-ML) and pBDL+pAL (pBDL+pAL-ML) mice. No significant differences according to Kruskal Wallis test (ANOVA on ranks) with Dunn's correction (A) or one-way ANOVA (B) with Tukey correction for multiple comparisons. The median + 95% CI is shown; control: n = 6, v-pBDL: n = 6, pBDL+pAL: n = 6 animals.
(DOCX)

**S9 Fig. Direct and indirect bilirubin.** Direct (A) and indirect bilirubin (B) in the blood plasma of mice, 14 days after v-pBDL or pBDL+pAL. No significant differences according to unpaired t-test. The median + 95% CI is shown; v-pBDL: n = 6, pBDL+pAL: n = 6 animals.
(DOCX)

**S10 Fig. Images from distinct stages during surgical interventions.** Identification of the common bile duct (black arrow in A). For cBDL: The common bile duct (black arrow) is separated from the proper hepatic artery and the portal vein (B) and 3-fold ligated and intersected (C). For v-pBDL: ligation of the left hepatic bile duct (black arrow) without damaging accompanying artery and vein (D).
(DOCX)

**S11 Fig. Experimental setup.** At the beginning of the experiment, the mice were allowed to practice burrowing two times. Afterward, body weight (bw), the distress score (ds), and burrowing activity (b) were measured on the indicated days. On day 0, the body weight of the animals was assessed and afterwards, the bile duct ligation (BDL), either cBDL, pBDL, v-pBDL or pBDL + pAL, was performed. The blood and tissue samples were collected on day 14. Data presented in this manuscript during the pre-, early, middle, or late phase of the experiment are labelled in blue print. Please note that we present percentage change in body weight and change in burrowing activity in this mansucript (the reference data, which were used for this calculation are presented in green print). Please also note that reduced food uptake in response to a certain distress level takes one day to lead to a reduction in body weight. Therefore, body weight is measured one day after we measured the distress score.
(DOCX)

## Acknowledgments

The authors express their gratitude for the exceptional technical support provided by Berit Blendow, Eva Lorbeer and Dorothea Frenz. We also acknowledge ChatGPT 3.5, which was used to improve clarity, grammar, and syntax of some sentences written for this manuscript.

## Author Contributions

**Conceptualization:** Guanglin Tang, Dietmar Zechner.

**Data curation:** Guanglin Tang, Wiebke-Felicitas Nierath, Emily Leitner, Wentao Xie, Nico Seume, Xianbin Zhang, Dietmar Zechner.

**Formal analysis:** Guanglin Tang, Wiebke-Felicitas Nierath, Emily Leitner, Xianbin Zhang, Luise Ehlers, Dietmar Zechner.

**Funding acquisition:** Brigitte Vollmar, Dietmar Zechner.

**Investigation:** Guanglin Tang, Wiebke-Felicitas Nierath, Emily Leitner, Wentao Xie, Denis Revskij, Nico Seume, Dietmar Zechner.

**Project administration:** Dietmar Zechner.

**Resources:** Dietmar Zechner.

**Supervision:** Brigitte Vollmar, Dietmar Zechner.

**Validation:** Dietmar Zechner.

**Writing – original draft:** Guanglin Tang, Wiebke-Felicitas Nierath, Emily Leitner, Xianbin Zhang, Dietmar Zechner.

**Writing – review & editing:** Wentao Xie, Denis Revskij, Nico Seume, Luise Ehlers, Brigitte Vollmar.

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
