## [Decision Letter · Decision Letter 0]

23 Nov 2023

PONE-D-23-30237Comparing animal well-being between bile duct ligation modelsPLOS ONE

Dear Dr. Zechner,

Thank you for submitting your manuscript to PLOS ONE. After careful consideration, we feel that it has merit but does not fully meet PLOS ONE’s publication criteria as it currently stands. Therefore, we invite you to submit a revised version of the manuscript that addresses the points raised during the review process.

We look forward to receiving your revised manuscript.

Kind regards,

Muhammad Salman Bashir, M.S.C

Academic Editor

PLOS ONE

“GT, WFN, WX, EL, DZ and BV were supported by the Deutsche Forschungsgemeinschaft (DFG research group FOR 2591, ZE 712/1-1, ZE 712/1-2, VO 450/15-1 and VO 450/15-2). LE and DR received funding through the research project “EnErGie” by the European Social Fund (ESF; reference: ESF/14-BM-A55-007/18) and the Ministry of Education, Science, and Culture of Mecklenburg-Vorpommern.”

“The authors are grateful for the perfect technical assistance from Berit Blendow and Dorothea Frenz. This study was supported by the Deutsche Forschungsgemeinschaft (DFG research group FOR 2591, ZE 712/1-1, ZE 712/1-2, VO 450/15-1 and VO 450/15-2). LE and DR received funding through the research project “EnErGie” by the European Social Fund (ESF; reference: ESF/14-BM-A55-007/18) and the Ministry of Education, Science, and Culture of Mecklenburg-Vorpommern.”

“GT, WFN, WX, EL, DZ and BV were supported by the Deutsche Forschungsgemeinschaft (DFG research group FOR 2591, ZE 712/1-1, ZE 712/1-2, VO 450/15-1 and VO 450/15-2). LE and DR received funding through the research project “EnErGie” by the European Social Fund (ESF; reference: ESF/14-BM-A55-007/18) and the Ministry of Education, Science, and Culture of Mecklenburg-Vorpommern.”

Reviewers' comments:

Reviewer's Responses to Questions

**Comments to the Author**

1. Is the manuscript technically sound, and do the data support the conclusions?

Reviewer #1: Partly

Reviewer #2: Yes

2. Has the statistical analysis been performed appropriately and rigorously? 

Reviewer #1: Yes

Reviewer #2: I Don't Know

3. Have the authors made all data underlying the findings in their manuscript fully available?

Reviewer #1: Yes

Reviewer #2: Yes

4. Is the manuscript presented in an intelligible fashion and written in standard English?

Reviewer #1: Yes

Reviewer #2: Yes

5. Review Comments to the Author

Reviewer #1: In the manuscript "Comparing Animal Well-being between Bile Duct Ligation Models", Guanglin Tang et al. compared the differences between cBDL and pBDL mice. They found that pBDL animals had a significantly higher survival rate and their well-being was significantly improved when compared to cBDL animals. After 14 days of ligation, liver enzymes such as aspartate and alanine aminotransferase, alkaline phosphatase, and glutamate dehydrogenase were significantly elevated after cBDL, but only glutamate dehydrogenase activity was increased after pBDL. However, several concerns regarding the research must be clarified.

(1) The authors should provide serum levels of TBA (total bile acids), TBIL (total bilirubin), IBIL (indirect bilirubin), and DBIL (direct bilirubin). These markers are commonly used to assess the degree of cholestasis in animals.

(2) The authors should provide inflammatory and fibrotic markers from cBDL and pBDL mice. Inflammation and fibrosis are common pathophysiological processes in biliary obstruction, and analysis of these markers will further support the differences between the two models.

(3) The authors should provide data on the levels of Mrp2, Mrp3, Mrp4, Bsep, Ntcp, Cyp7a1, Cyp8b1, and Cyp27a1 mRNA in total liver tissue. This is important because these genes encode enzymes involved in bile acid synthesis and transport, and their expression levels can provide insights into the extent of cholestasis and liver injury in the two models. By comparing the expression levels of these genes between cBDL and pBDL mice, the authors can further validate their hypothesis that pBDL results in less severe liver damage compared to cBDL. Therefore, it is essential to include this information in the manuscript to strengthen the study's conclusions.

(4) This study may have some value in certain aspects, but overall, it lacks innovation.

Reviewer #2: The common bile duct ligation (cBDL) is the most used animal model to mimic hepatic diseases such as obstructive cholestasis, primary biliary or sclerosing cholangitis and acute liver injury. Recently, it has been introduced a modification in this model, just the left hepatic bile duct is ligated creating a partial bile duct ligation (pBDL). Within the left lobe pBDL causes similar conditions of cBDL like necrosis, inflammation and fibrosis. The Authors aimed to compare animal wellbeing between cBDL and pBDL mice, demostrating that mice after pBDL show less distress and a better survival than mice after cBDL. The manuscipt is interesting but some points need to be improved:

- Regarding the morphological study of the heaptic tissue, in the comparison of liver necrosis and fibrosis (Figure 6), it is important to add pictures and data of cBDL and unligated lobe or control tissue. In general, they should insert the study even of the right lobe to exclude any type of damage in that side compared the left one.

- In the same Figure 6, the images of sirius red are very out of focus, requiring higher contrast to better appreciate the differences between the various samples.

- The Discussion section needs to be rewrite, in the present form is full of repetitive concepts. The Authors should semplify it by discussing and speculating the points presented in a gradual and progressive way.

- It is not completely clear the reason why the pBDL animal model could be used to study local features of the mentioned hepatic diseases. Please, clarify better this point in the Discussion section.

- When they observed that the necrosis in pBDL + pAL animal model is caused by interrupting of arterial blood supply and, at the same time, induce cholestasis, they should insert the important role of the Peribiliary Plexus (PBP) which supply bile ducts originating from the hepatic artery.

- Since the crucial point of the study is the ligation of the left hepatic bile duct, it would be really important to add pictures of the surgical procedure.

- Several typos are present along the text, please revise it carefully.

6. PLOS authors have the option to publish the peer review history of their article (what does this mean?). If published, this will include your full peer review and any attached files.

Reviewer #1: No

Reviewer #2: No

---

## [Author Response · Author response to Decision Letter 0]

22 Apr 2024

Reply to reviewer 1

Thank you very much for all of your valuable suggestions to improve the manuscript. We have followed all of your recommendations and incorporated a significant amount of additional data. This process has expanded the number of figures and supplemental figures from 9 in the original manuscript to 20 in the current version. In order to effectively convey the main messages of our study, we restructured the results section of our manuscript into three blocks. We now compare CBDL to pBDL in Fig 1 and S1 Fig. We compare cBDL to the v-pBDL animal model in SFig 2-4 and Fig 2-4. And we compare the v-pBDL to the pBDL+pAL animal model (both ligated as well as unligated liver lobes) in SFig 5-9 and Fig 5-9. I hope that these additional experiments are now sufficient for acceptance as a publication in PLOS ONE. For a point to point reply please see text below.

Point 1 from Reviewer 1: The authors should provide serum levels of TBA (total bile acids), TBIL (total bilirubin), IBIL (indirect bilirubin), and DBIL (direct bilirubin). These markers are commonly used to assess the degree of cholestasis in animals.

Answer: For comparing cBDL to the v-pBDL animal model we now present TBA (total bile acids) and TBIL (total bilirubin) in figure 3E and figure 3F and IBIL (indirect bilirubin), and DBIL (direct bilirubin) in supplemental figure 4. For comparing v-pBDL to pBDL+pAL we now present TBA (total bile acids) and TBIL (total bilirubin) in figure 7E and figure 7F and IBIL (indirect bilirubin), and DBIL (direct bilirubin) in supplemental figure 9.

Point 2 from Reviewer 1: The authors should provide inflammatory and fibrotic markers from cBDL and pBDL mice. Inflammation and fibrosis are common pathophysiological processes in biliary obstruction, and analysis of these markers will further support the differences between the two models.

Answer: For comparing cBDL to the v-pBDL animal model we now present fibrotic (collagen I deposition) and inflammatory markers (CAE+ cells and TNFa as well as IFNg expression) in figure 2E-L, and supplementary figure 3. For comparing the ligated liver lobes of v-pBDL to pBDL+pAL nice we now present fibrotic (collagen I deposition) and inflammatory markers (CAE+ cells and TNFa as well as IFNg expression) in figure 6D-I, and supplementary figure 7. For comparing the unligated liver lobes of v-pBDL to pBDL+pAL mice we now present fibrotic (collagen I deposition) and inflammatory markers (CAE+ cells and TNFa as well as IFNg expression) in supplemental figure 6D-I, and supplementary figure 8.

Point 3 from Reviewer 1: The authors should provide data on the levels of Mrp2, Mrp3, Mrp4, Bsep, Ntcp, Cyp7a1, Cyp8b1, and Cyp27a1 mRNA in total liver tissue. This is important because these genes encode enzymes involved in bile acid synthesis and transport, and their expression levels can provide insights into the extent of cholestasis and liver injury in the two models. By comparing the expression levels of these genes between cBDL and pBDL mice, the authors can further validate their hypothesis that pBDL results in less severe liver damage compared to cBDL. Therefore, it is essential to include this information in the manuscript to strengthen the study's conclusions.

Answer: For comparing cBDL to the v-pBDL animal model we now present data on Mrp2, Mrp3, Mrp4, Bsep, Ntcp, Cyp7a1, Cyp8b1, and Cyp27a1 mRNA in total liver tissue in figure 4A-H. For comparing the ligated liver lobes of v-pBDL to pBDL+pAL nice we now present data on the expression of these genes in figure 8A-H. For comparing the unligated liver lobes of v-pBDL to pBDL+pAL mice we now present the expression of these genes in supplemental figure 9A-H.

Point 4 from Reviewer 1: This study may have some value in certain aspects, but overall, it lacks innovation.

Answer: We had already the following completely novel aspects in the original publication: We could demonstrate that pBDL causes significant less distress than cBDL and leads to a higher survival rate of the animals. In addition, we described for the first time pathological consequences when ligating the left bile duct with the corresponding left hepatic artery. In the revised manuscript we added a more detailed comparison between the three animal models, cBDL, pBDL and pBDL+pAL (the pBDL+pAL model is completely novel) offering among other things gene expression regulation of bile acid synthesizing enzymes and bile acid transporter proteins (this is to our knowledge completely novel for the pBDL and pBDL+pAL animal models).

Reply to reviewer 2

Thank you very much for all of your valuable suggestions to improve the manuscript. We have followed all of your recommendations and incorporated a significant amount of additional data. This process has expanded the number of figures and supplemental figures from 9 in the original manuscript to 20 in the current version. In order to effectively convey the main messages of our study, we restructured the results section of our manuscript into three blocks. We now compare CBDL to pBDL in Fig 1 and S1 Fig. We compare cBDL to the v-pBDL animal model in SFig 2-4 and Fig 2-4. And we compare the v-pBDL to the pBDL+pAL animal model (both ligated as well as unligated liver lobes) in SFig 5-9 and Fig 5-9. I hope that these additional experiments are now sufficient for acceptance as a publication in PLOS ONE. For a point to point reply please see text below.

Point 1 from Reviewer 2: Regarding the morphological study of the hepatic tissue, in the comparison of liver necrosis and fibrosis (Figure 6), it is important to add pictures and data of cBDL and unligated lobe or control tissue. In general, they should insert the study even of the right lobe to exclude any type of damage in that side compared the left one.

Answer: For comparing cBDL to the v-pBDL animal model we now present pictures and data on liver necrosis, fibrosis and CAE staining in figure 2A-L. For comparing the ligated liver lobes of v-pBDL to pBDL+pAL nice we now present pictures and data on liver necrosis, fibrosis and CAE staining in figure 6A-I. For comparing the unligated liver lobes of v-pBDL to pBDL+pAL mice we now present pictures and data on liver necrosis, fibrosis and CAE staining in supplemental figure 6A-I.

Point 2 from Reviewer 2: In the same Figure 6, the images of sirius red are very out of focus, requiring higher contrast to better appreciate the differences between the various samples.

Answer: Those images, in my opinion were in focus (however a limitation might be that my eyesight has decreased over time), but the histology was hard to judge, because we did not counterstain the tissue with hematoxylin. In order to make it easier to judge the tissue, we did a collagen I immunohistochemistry and counterstained the tissue with hematoxylin. These images are now presented in figure 2E-G (comparing healthy control liver to liver sections from cBDL and v-pBDL mice), figure 6D-E (comparing the ligated liver lobes of v-pBDL to pBDL+pAL) and , supplemental figure 6D-E (comparing the unligated liver lobes of v-pBDL to pBDL+pAL).

Point 3 from Reviewer 2: The Discussion section needs to be rewrite, in the present form is full of repetitive concepts. The Authors should semplify it by discussing and speculating the points presented in a gradual and progressive way.

Answer: We shortened, restructured and rewrote the discussion section (line 350-403 and line 444-457). However, we also added a section about the expression of genes involved in in bile acid synthesis and transport, because analyzing those genes was requested by another reviewer (line 404-443).

Point 4 from Reviewer 2: It is not completely clear the reason why the pBDL animal model could be used to study local features of the mentioned hepatic diseases. Please, clarify better this point in the Discussion section.

Answer: we now clarified (by shortening and restructuring the text) the point that the same local pathophysiological features as observed in the mentioned hepatic diseases can be studied after pBDL (see line 378-386)

Point 5 from Reviewer 2: When they observed that the necrosis in pBDL + pAL animal model is caused by interrupting of arterial blood supply and, at the same time, induce cholestasis, they should insert the important role of the Peribiliary Plexus (PBP) which supply bile ducts originating from the hepatic artery.

Answer: we inserted the important role of the peripiliary venous plexus when we discussing pBDL + pAL mice (see line 395-399)

Point 6 from Reviewer 2: Since the crucial point of the study is the ligation of the left hepatic bile duct, it would be really important to add pictures of the surgical procedure.

Answer: We added pictures as supplemental figure 10 (S10 Fig).

Point 7 from Reviewer 2: Several typos are present along the text, please revise it carefully.

Answer: We corrected the typos throughout the manuscript.

---

## [Editor Report · Decision Letter 1]

1 May 2024

Comparing animal well-being between bile duct ligation models

PONE-D-23-30237R1

Dear Dr. Dietmar,

We’re pleased to inform you that your manuscript has been judged scientifically suitable for publication and will be formally accepted for publication once it meets all outstanding technical requirements.

Kind regards,

Muhammad Salman Bashir, M.S.C

Academic Editor

PLOS ONE
---

## [Editor Report · Acceptance letter]

28 May 2024

PONE-D-23-30237R1 

PLOS ONE

Dear Dr. Zechner, 

I'm pleased to inform you that your manuscript has been deemed suitable for publication in PLOS ONE. Congratulations! Your manuscript is now being handed over to our production team.

Kind regards, 

on behalf of

Dr. Muhammad Salman Bashir 

Academic Editor

PLOS ONE